# Effect of Acupuncture on Functional Capacity in Patients Undergoing Hemodialysis: A Patient-Assessor Blinded Randomized Controlled Trial

**DOI:** 10.3390/healthcare10101947

**Published:** 2022-10-05

**Authors:** Marta Correia de Carvalho, José Nunes de Azevedo, Pedro Azevedo, Carlos Pires, Manuel Laranjeira, Jorge Pereira Machado

**Affiliations:** 1ICBAS–School of Medicine and Biomedical Sciences, University of Porto, 4050-313 Porto, Portugal; 2TECSAM-Tecnologia e Serviços Médicos SA, 5370-530 Mirandela, Portugal; 3Center for Research in Neuropsychology and Cognitive and Behavioral Intervention (CINEICC), Faculty of Psychology and Educational Sciences, University of Coimbra, 3000-115 Coimbra, Portugal; 4INC–Instituto de Neurociências, 4100-141 Porto, Portugal; 5CBSin—Center of BioSciences in Integrative Health, 4000-105 Porto, Portugal

**Keywords:** chronic kidney disease, hemodialysis, acupuncture, functional capacity, integrative medicine

## Abstract

Decreased functional capacity (FC) in patients undergoing hemodialysis (HD) is associated with adverse health events and poor survival. Acupuncture is recognized as a safe and effective integrative treatment. The aim of this study is to evaluate the effect of acupuncture treatment on the FC in chronic kidney disease with GFR category 5 (CKG G5) patients undergoing HD. In this patient-assessor blinded randomized controlled trial, seventy-two KF patients were randomly assigned to experimental (*n* = 24), placebo (*n* = 24) and control groups (*n* = 24). The primary outcome was the improvement in FC assessed by the 6-Minute Walk Test (6-MWT). Secondary outcomes included assessment of peripheral muscle strength by the Handgrip Strength Test (HGS) and the 30-Second Sit-to-Stand Test (STS-30) at baseline, after treatment and at 12-week follow up. A mixed ANOVA with interaction time*group was used. The experimental group increased walk distance (*p* < 0.001), lower limbs strength (*p* < 0.001) and handgrip strength (*p* = 0.012) after nine acupuncture sessions and stabilized in the follow-up (*p* > 0.05). In the placebo and control groups the 6-MWT and 30STS results decreased (*p* < 0.001) and the HGS scores did not change through time (*p* > 0.05). Acupuncture treatment improved FC and muscle strength in patients undergoing HD.

## 1. Introduction

The growing incidence and prevalence of chronic kidney disease (CKD) are associated with increased morbidity, mortality and additional burden and economic cost to the healthcare system and social assistance [1,2]. According to the most recent data, in 2016, CKD was ranked as the 16th leading cause of death worldwide, and by 2040 it is expected to be the fifth leading cause of death [3].

The etiology of CKD is diverse, with diabetes and hypertension mainly contributing to the increased prevalence of the disease [4]. Patients who have CKD with glomerular filtration rate (GFR) category 5 (CKG G5) can only be treated by kidney replacement therapy (KRT), in particular peritoneal dialysis (PD), hemodialysis (HD), kidney transplantation (KT) and conservative kidney management [4]. Technological innovation and optimization of medical therapy applied to hemodialysis treatments contributed to patients surviving longer on dialysis and after kidney transplantation, helping to increase worldwide the prevalence of kidney failure [5,6,7].

HD is the most predominant dialysis modality and is characterized as a complex and demanding treatment regimen. One of the significant problems in patients on maintenance HD and a substantial predictor of morbidity and mortality is skeletal muscle atrophy, which contributes significantly to reduced functional capacity [8,9].

Even though traditional Chinese medicine (TCM) is based on different concepts of etiopathogenesis, semiology, physiology and therapeutics, it is being widely accepted by Western medicine. Currently, there is increasing use of complementary and integrative therapies with adjunctive interventions to manage symptoms arising from CKD [10,11]. The application of TCM and acupuncture concepts in the field of nephrology has been increasingly described in recent research [12]. Kim et al. (2011) studied patients undergoing hemodialysis receiving acupuncture treatment for some of their symptoms, and acupuncture seemed to be feasible and safe for symptom management [13]; Bullen et al. (2018) have concluded that acupuncture during HD may contribute toward the improvement of health-related QoL [14]; Yu et al. (2017) evaluated the feasibility effect of acupuncture on renal function in patients with CKD and results suggest reduced creatinine levels and increased GFR [15].

Despite the increasing number of RCT and meta-analysis articles, more scientific research is needed to validate the therapeutic effects of acupuncture. A systematic review suggests that future high-quality randomized controlled trials are required to verify the safety and effectiveness in pre-dialysis patients with CKD and those undergoing dialysis, to compare acupuncture with placebo and verify whether intensive short-term acupuncture interventions or ongoing but less frequent treatments are advantageous for patients [16].

According to the literature review and the best of the authors’ knowledge, no similar study was found assessing the effect of acupuncture or treatment frequency on improving FC and muscle strength in HD patients. Several studies have focused on the symptomatic treatment of other functional state. Damasceno et al. (2019) analyzed the effect of manual acupuncture on muscle mass and inflammation markers in older adults with sarcopenia [17]; Zhou et al. (2012) conducted a pilot study to determine the effect of unilateral manual acupuncture on selected acupoints on ankle dorsiflexion strength of both limbs on young men [18]; Wang et al. (2020) evaluated whether three weekly acupuncture treatment sessions were superior to one weekly session for symptomatic outcomes in postprandial distress syndrome [19]; Lin et al.(2020) assessed the symptomatic improvement in patients with knee osteoarthritis on three sessions per week of acupuncture compared to one session per week [20]. The evidence for the role of acupuncture treatment frequency is unclear, and there are currently no defined and accepted consensus criteria for treatment frequency [21].

Given the paucity of scientific research on the clinical application of acupuncture in patients on kidney replacement therapy, the present study was designed to evaluate the effect of acupuncture on FC in patients with CKD G5 undergoing HD and to analyze the differences in the frequency of treatment (three times a week or once a week).

## 2. Materials and Methods

### 2.1. Design and Settings

This parallel group patient-assessor blinded randomized controlled trial (RCT) was conducted at the Hemodialysis Center of TECSAM—Tecnologia e Serviços Médicos, SA, in Mirandela, Portugal, from August 2021 to February 2022. The study protocol and informed consent were approved by the Ethics Committee of Centro Hospitalar Universitário do Porto/Instituto Ciências Biomédicas Abel Salazar (registered number 2019/CE/P026_P304/2019/CETI) and are an integral part of the research project of the Doctoral Program in Biomedical Sciences of ICBAS School of Medicine and Biomedical Sciences, University of Porto, Portugal. This trial was registered at ClinicalTrials.gov (NCT05362643).

### 2.2. Participants

Potentially eligible participants were identified by TECSAM’s nephrologist, and an advertisement was posted on the hemodialysis center notice board inviting patients to participate.

Inclusion and exclusion criteria were applied in the participant selection process. Male and female patients, age 18 years or older, who had been receiving regular HD treatment for more than 3 months, 3 sessions per week, 4 h per session, with the medically stable condition were included. Excluded were patients who refused to participate in the study, those with a clinical indication that prevented their participation in the research, and patients having other comorbidities such as poorly controlled malignant hypertension, unstable angina, uncontrolled diabetes mellitus, cerebrovascular failure with recurrent syncope, uncontrolled heart failure, severe mental illness or cognitive impairment, or inability to practice physical exercise. Patients that have had acupuncture treatment in the past two weeks, have known hypersensitivity reaction and/or other side effects after acupuncture treatment, inability to cooperate with the procedures inherent to the application of the procedure were also excluded.

After screening for eligibility, a written informed consent explaining the objectives and procedures for the research project implementation was obtained from everyone who agreed to participate in compliance with the revised version of the Declaration of Helsinki and the Oviedo Convention. This procedure was performed by an independent evaluator who assigned each an anonymous code.

### 2.3. Intervention

Our research team developed a standardized and repeatable protocol. The selection of acupuncture points was based on a literature review, on the general principles of acupuncture and traditional Chinese medicine (TCM) meridian theory, clinical experience, and consensus method by experts in TCM, acupuncture and nephrology [22,23,24,25].

Experimental (verum acupuncture) and placebo (sham acupuncture) groups received 9 acupuncture treatments. Both groups were divided into two subgroups with a different treatment frequency: subgroup A received 3 acupuncture sessions per week over 3 weeks; subgroup B received 1 acupuncture session per week over 9 weeks. The same selection of acupuncture points was applied to both experimental subgroups A and B. Both placebo subgroups A and B received acupuncture on the same non-acupuncture points.

Details of needling and treatment regimen according to the revised Standards for Reporting Interventions in Clinical Trials of Acupuncture (STRICTA)—2010 Checklist [26] are listed in Table 1.

Locations of the selected acupuncture and non-acupuncture points for this study are described in Appendix B, Table A1 and Table A2.

The acupuncture treatment was provided by a licensed specialist in TCM with professional card number C-006513 issued by the regulator of health system central administration in Portugal, ACSS-Administração Central do Sistema de Saúde, I.P., with five years of professional experience.

Apart from the usual care routine for hemodialysis sessions, no further interventions were allowed during the study period. During the HD session, acupuncture treatment took place at the Hemodialysis Center of TECSAM Tecnologia e Serviços Médicos, SA, in Mirandela, during the hemodialysis session.

### 2.4. Outcome Measurement

The primary outcomes were the change in functional capacity measured by the distance walked (meters) obtained by the 6-Minute Walk Test (6MWT) and whether there were differences in the frequency of treatment (three times a week or once a week).

Secondary outcomes included changes in peripheral muscle strength. Lower limb strength was measured by 30-Second Sit-to-Stand (STS-30) Test, and handgrip strength was measured by Handgrip Strength (HGS) Test.

All assessments were performed at baseline, after treatment and at a 12-week follow-up. Demographic, clinical and laboratory data were collected at baseline as described in the participant timeline in Appendix B, Table A3.

Regarding safety, and since the acupuncture treatment was performed during the period of the hemodialysis session, patients were under permanent medical surveillance, and any adverse event would be immediately communicated.

#### 2.4.1. Six-Minute Walk Test (6MWT)

The 6MWT assesses the submaximal level of functional exercise capacity and is a simple and convenient to use walking test. Changes in the six-minute walk distance are used to evaluate the efficacy of therapeutic interventions [27].

According to the “ATS Statement: Guidelines for the Six-Minute Walk Test” [27], patients were instructed to walk as fast as possible during 6 min on a 30 m straight, enclosed corridor. The corridor length was marked every 3 m, and the turnaround points were marked with an orange traffic cone. A colored tape was used to mark the starting line on the ground, indicating the beginning and end of each 60 m turn. The distance walked was registered in meters. Patients were allowed to stop and rest during the test. However, they were encouraged to resume walking as soon as possible.

#### 2.4.2. Thirty-Second Sit-to-Stand (STS-30) Test

The STS-30 test is a reliable test for functional evaluation in hemodialysis patients and is used to assess functional lower extremity strength [28,29].

Patients were asked to stand up and sit down again in a chair without the help of their arms, and to perform the most significant number of repetitions in 30 s, in a single attempt. For this, they adopted a posture with the trunk erect, arms crossed in front of the chest, and feet flat on the ground.

#### 2.4.3. Handgrip Strength (HGS) Test

Handgrip strength was measured, in kilograms-force (KGF), by the HGS Test [30,31], using an electronic hand dynamometer (CAMRY, EH101). The HGS test is reliable, simple to use and an indicator of overall muscle strength [32].

Patients were asked to sit in a chair, with their backs supported on the backrest, feet fully supported on the floor, with the extremity of the upper limb forming a 90° angle with the trunk, elbow fully extended, pressing as hard as possible on the dynamometer with finger flexion.

This test was repeated twice with each hand, with a one-minute interval between each repetition, and the best attempt of each hand was considered.

### 2.5. Randomization and Blinding

Patients were randomly allocated in a 1:1:1 ratio and assigned to an experimental group (verum acupuncture), placebo group (sham acupuncture), or control group (waiting list) following simple randomization procedures through a computerized random number list. Randomization sequences were created using Microsoft^®^ Excel^®^ for Microsoft 365 MSO, Microsoft, Washington, USA, by an independent researcher who prepared the assignments in opaque envelopes and took charge of the allocation sequence concealment.

After the random assignment of the participants to the three groups and regarding the frequency of treatment, it was defined that the first 12 randomized patients in the VA and SA groups would receive treatment 3 times a week, and the remaining 12 would receive treatment once a week.

Participants, outcome assessor and statistician were blind to the allocation. Regarding the intervention and its specific nature, only the traditional Chinese medicine practitioner was not blind to the verum acupuncture, sham or non-acupuncture groups.

### 2.6. Quality Control and Data Collection

Before the clinical trial began, the research team received specialized training to maintain the study quality. Topics such as research process, eligibility criteria, data collection and management, and adverse events assessment were covered.

Screening for eligibility, demographic and clinical data collection, and physical examination were performed by the TECSAM clinical team as an independent assessor. The leading researcher, who provided the acupuncture treatments, had no access to patient data until anonymization, randomization and allocation procedures were complete. Baseline, after-treatment and follow-up outcome variables were assessed by the TECSAM head nurse, who did not know which group each subject was assigned to. Additionally, the statistician was unaware of the allocation during the research period to guarantee the objectivity and impartiality of the data.

### 2.7. Statistical Analysis and Sample Size

Statistical analysis was performed with IBM SPSS Statistics software, version 27.0, IBM, New York, USA [33].

Demographic, clinical and laboratorial data at baseline were described by group through the mean, standard deviation and frequencies. Groups were compared with Fishers’ Exact Test (categorical variables) and the one-way ANOVA (continuous variables). A mixed ANOVA with interaction time*group and time*regimen was used to assess the acupuncture’s effect and the frequency of treatment, respectively. As the interaction time*treatment frequency was not significant, a new mixed ANOVA was performed without the factor treatment frequency. Differences through time within each group were assessed with the repeated-measures ANOVA. The Bonferroni correction was used for multiple comparisons. Partial Eta squared (η^2^p) was used to assess the effect size of the interaction, considering the thresholds: η^2^p = 0.01 small effect, η^2^p = 0.06 medium effect, η^2^p = 0.14 large effect [34]. The assumptions for the mixed ANOVA were assessed and validated through Shapiro–Wilk’s Test (normality), Levene’s Test (homogeneity of variances), and Box’s M Test (sphericity of the covariance matrix). A significance level of 5% was considered for the statistical tests.

To assess the effectiveness of blinding, participants of verum acupuncture (VA) and sham acupuncture (SA) groups were asked, after treatment, to guess what type of acupuncture treatment they believe they had received (“Verum”, “Sham”, “I do not know”). The answers were used to calculate the James Blinding Index (BI). This index ranges from 0 (lack of blinding) to 1 (complete blinding); values above 0.50 indicate that the subjects have been blinded [35,36]. To calculate the BI, the package “BI” available in the statistical software R was used [37].

Owing to the lack of previous data, the sample size was estimated to achieve a small to medium effect size (f = 0.19) [34] in the mixed ANOVA (3 groups × 3 repeated measures) with statistical power of 80% and a significant level of 5%. The calculations, carried out with G*Power V.3.1, Heinrich Heine University Dusseldorf, Dusseldorf, Germany [38], led to estimate a minimum of 20 patients in each group (Acupuncture Group, Sham Acupuncture Group and Waiting-List Group). Computational formulas for sample size estimation can be found in the paper that describes the software G*Power [38].

## 3. Results

From December 2020 to May 2021, 88 HD patients from the TECSAM Hemodialysis center were assessed for eligibility; 10 were excluded because they did not meet the inclusion criteria, and 6 declined to participate. A total of 72 patients were included and randomly assigned to the study groups: 24 in the verum acupuncture (VA) group, 24 in the sham acupuncture (SA) group, and 24 in the waiting-list (WL) group.

During the study period, between the post-treatment assessment and before the follow-up assessment, 1 patient in the VA group, 2 in the SA group and 2 in the WL group dropped with the causes identified in Figure 1. Therefore, 67 HD patients were included in the complete analysis set, as shown in the study flow diagram (Figure 1).

The sample included 67 hemodialysis patients aged between 56 and 91, with a mean age of 71.6 (SD = 7.7). Most were men (61.2%), lived in rural areas (64.2%), were retired (85.1%), and had low education levels (73.1% with the 1° Cycle and 7.5% without any education level). No statistical differences were found between the groups in any of these variables (*p* > 0.05). As for the clinical characteristics, the most prevalent cause of CKD was diabetes mellitus (43.3%). Most of the patients were undergoing HD treatment for over a year and less than 10 years (83.6%) and used arteriovenous fistula (91.0%). No statistical differences were found between the groups regarding the clinical and laboratory variables (*p* > 0.05).

The baseline demographic, clinical and laboratorial characteristics are shown in Table 2.

Table 3 shows the results of the ANOVA with interaction time*group*treatment frequency for 6MWT, STS-30 Test and HGS Test. The interaction time*group was statistically significant for total walked distance measured by the 6MW Test (*p* < 0.001, η^2^p = 0.521), for lower limb strength measured by the STS-30 Test (*p* < 0.001, η^2^p = 0.600), and for handgrip strength measured by the HGS Test (*p* = 0.032, η^2^p = 0.083). These results lead to conclude that the evolution through time differed significantly amongst the groups (Table 3).

As for the interaction time*treatment frequency, no statistically significant effects were found for the results of 6MWT (*p* = 0.172, η^2^p = 0.028), STS-30 Test (*p* = 0.438, η^2^p = 0.027) or HGS Test (*p* = 0.843, η^2^p = 0.002), showing that the evolution through time did not differ significantly between acupuncture treatment frequency (3 times/week × 3 weeks vs. 1 time/week × 9 weeks) (Table 3).

Owing to the absence of significant differences between treatment frequencies, a new ANOVA was performed without the factor treatment frequency. Results of the new ANOVA with interaction time*group for each variable are presented in Table 4 and Figure 2.

Results show that the evolution through time of the three variables differed significantly among the groups (*p* < 0.05 for the interaction time*group). The effects were large for 6MWT (*p* < 0.001, η^2^p = 0.509) and the STS-30 Test (*p* < 0.001, η^2^p = 0.565), and medium for the HGS Test (*p* = 0.043, η^2^p = 0.083) (Table 4).

The analysis of the evolution within each group shows that, in the VA group, walked distance increased significantly (*p* < 0.05) from 307.2 ± 80.1 in the baseline assessment to 383.2 ± 70.7 after treatment, but did not change in the follow-up (368.9 ± 68.2) (*p* > 0.05). The mean in the follow-up was still significantly higher than the mean in the baseline assessment (*p* < 0.05). On the other hand, in the SA group and in the WL group, a trend of decrease in the walked distance was observed, with the mean in the follow-up moment significantly lower than the mean in the baseline assessment (*p* < 0.05) (Table 4 and Figure 2).

Similar results were found in the lower limb strength — the mean of the VA group increased from 9.7 ± 2.8 in the baseline assessment to 11.5 ± 2.8 after treatment (*p* < 0.05) but did not change in the follow-up (11.2 ± 3.0) (*p* > 0.05). The mean in the follow-up was significantly higher than the mean in the baseline assessment (*p* < 0.05). The means in the SA group and in the WL group decreased significantly through time (*p* < 0.05) (Table 4 and Figure 2).

As for the isometric handgrip force, in the VA group, the mean increased from 21.4 ± 5.5 in the baseline assessment to 22.8 ± 5.7 after treatment (*p* < 0.05) but did not change significantly in the follow-up (21.7 ± 5.1) (*p* > 0.05). There were no significant differences between the means in the baseline assessment and the follow-up in this group (*p* > 0.05). No significant differences through time were found in the SA group (*p* = 0.601) or in the WL group (*p* = 0.112) (Table 4 and Figure 2).

Results of the blinding success assessment (Table 5) show that 31.1% of the participants believe that they received verum acupuncture (30.4% in VA and 31.8% in SA) and 68.9% answered “I do not know” (69.6% in VA and 68.2% in SA)—none of the patients answered “sham acupuncture”. The value of the Blinding Index of 0.84 (95% confidence interval: 0.78–0.91) allows for concluding that the participant blinding was successful.

Throughout the trial period, no adverse events related to treatment were observed or reported by patients, practitioner or care providers, suggesting the safety of using acupuncture during hemodialysis sessions.

## 4. Discussion

CKD is a debilitating disease and as it progresses to kidney replacement therapy, patients often experience numerous symptoms that negatively impact their quality of life, affecting their physical, functional, metabolic, social and mental conditions with high impact on functional capacity when compared to healthy individuals [8,39]. Although there are only a few clinical trials comparing the effect of different acupuncture treatment frequencies, an appropriate dose of acupuncture must be considered in order to improve outcomes [21].

In this study, a parallel patient-assessor blinded RCT was conducted to evaluate the effect of acupuncture in patients with kidney failure undergoing HD and also to establish the differences in the frequency of treatment (three times a week or once a week). The outcomes showed that nine sessions of manual acupuncture improved functional capacity and increased peripheral muscle strength in the experimental group (VA group) compared to the placebo (SA) and control (WL) groups.

The primary and secondary outcomes did not depend on the frequency of acupuncture treatment since no significant differences were observed, indicating that three treatments per week do not provide better results than one treatment per week on functional capacity and muscle strength in patients undergoing HD.

To the authors’ knowledge, no RCT was performed to assess the effect of different acupuncture treatment frequencies on FC parameters and in the clinical sample of the present study. The impact of the frequency of acupuncture has been investigated in other medical conditions, and the reflections found are not consensual. The pilot study conducted by Yuan et al. (2009), designed to compare the effectiveness of two different frequencies, two versus five times per week of acupuncture treatment for chronic low back pain, showed no significant differences among groups in any outcome, and at each time point [40]. Lin et al. (2020) reported that three sessions per week of acupuncture produced superior and persisting improvement (≥16 weeks) in knee osteoarthritis treatment compared with one session per week of acupuncture [20]. Wang et al. (2020) conducted a pilot study to establish if three sessions per week of acupuncture treatment were superior to one session per week for symptomatic outcomes in postprandial distress syndrome. After four weeks of treatment, three sessions per week tended to improve symptoms and quality of life compared to once a week [19]. Given the results of the present study, it cannot be said that more frequent weekly acupuncture treatments provided greater therapeutic effects. Only studies in chronic pain seem to support this concept, which needs further research with more robust results.

Acupuncture treatment increased walked distance after treatment and lasted beyond a 12-week follow-up. The results of sham acupuncture (SA group) or no-acupuncture treatment (WL group) did not improve walked distance after treatment, followed by a decreasing tendency from baseline to 12-week follow-up. The longer lasting results of acupuncture treatment may offer the advantage to the patient to keep a greater functional capacity through time. Similar results were obtained for muscle strength after treatment. Verum acupuncture increased muscle strength, but it did not last beyond the 12-week follow-up period. A decrease in muscle strength was observed from baseline to 12-week follow-up in the SA and WL groups. Acupuncture also increased handgrip strength, at the end of treatment. Given the high percentage of elderly hemodialysis patients, many of whom need to walk with the help of a cane or a walker, the greater handgrip strength achieved with acupuncture treatment may offer the advantage of stronger support for their ambulation. At the 12-week follow-up, no significant differences were observed compared to baseline and post-treatment. No increase was found in grip strength in either the SA group or the WL group at different assessment time points.

No studies were found assessing the effectiveness of acupuncture in improving functional capacity and muscle strength in hemodialysis patients. Zhou et al. (2012) concluded that unilateral manual acupuncture and electroacupuncture at selected acupoints on ankle dorsiflexion could improve muscle strength in both limbs, and electroacupuncture at the non-acupoints as used in this study can also induce similar strength gains [18]. Hu et al. (2015) demonstrated the improvement of muscle atrophy in 5/6 nephrectomy rats by low-frequency electrical stimulation of an acupuncture point, due to increased protein metabolism and myogenesis [41]. Although these reports point to a positive effect of acupuncture, the study conducted by Soares Mendes Damasceno et al. (2019) showed no statistically significant difference between acupuncture and non-acupuncture groups on muscle mass, muscle strength, functionality, and inflammatory markers in older people with sarcopenia [17]. Perhaps these discrepancies are due to the fact that each of the studies had a specific acupuncture treatment protocol for the clinical conditions under investigation with different types of acupuncture, duration, treatment frequency and control interventions, so it is not possible to compare the results.

Regarding the results of the present study, it can be speculated that the improvement in functional capacity and muscle strength was due to the effect of acupuncture needle stimulation on increased blood flow, cytokines and growth factors that may result in modulation of sensory input afferents through changes in the connective tissue milieu [42].

In kidney failure patients undergoing HD, the 6MWT distance is considered an independent predictor of mortality. According to Kohl et al. (2012), the survival rate of patients undergoing HD increased by approximately 5% for every 100 m walked in the 6MWT [43]. Additionally, different studies have shown that handgrip strength predicts all-cause mortality in men and women on maintenance HD [44,45,46].

The results of this study are encouraging and may contribute to the integration of acupuncture as an integrative therapy for symptom management in HD patients. The progression of CKD and the initiation of dialysis are associated with adverse outcomes or increased mortality. An increase in functional capacity and muscle strength means that HD patients will be able to perform activities of daily living more autonomously and with a higher level of functioning.

Strengths of the present study include comparing the experimental group with two groups (sham and non-acupuncture groups); the participants, the outcome evaluator and the statistician were blinded to allocation; and the number of dropouts during follow-up was as low as possible.

Patients undergoing HD have to cope with a stressful and disruptive chronic illness and suffer from emotional problems that negatively impact on their overall health [7,47]. Given this assumption and the fundamentals of TCM, the acupuncture points *Shenmen* (HT7) and *Guanyuan* (CV4) were included in the study protocol. The *Shenmen* (HT7) acupoint was chosen since it is an important acupuncture point for calming and regulating the spirit. The *Guanyuan* (CV4) acupuncture point is described as a crucial point for strengthening the body and mind. Moreover, as it tonifies the kidney and the original Qi, it is recognized as an effective point in the case of chronic diseases [22].

Despite the methodological rigor considered in the design of this study, a few limitations can be found: first, the sample size used to compare different frequencies of acupuncture treatment, which resulted from the subdivision of the participants from experimental and placebo groups; second, the TCM practitioner who provided acupuncture treatment cannot be blinded due to the type of intervention; finally, although the effect of acupuncture treatment had been observed 12 weeks after the end of treatment, this time interval was not long enough to evaluate the long-term effect of acupuncture.

Further research is required to verify the tested acupuncture treatment protocol and validate its clinical efficacy in a significant sample of patients with CKD without kidney replacement therapy or in a healthy population. Complete data regarding nutritional status might be valuable to correlate with the effect of acupuncture on the parameters evaluated by this RCT. In addition, an active control group should be considered in the design of a future study, such as a comparison with an intradialytic exercise or physical therapy group.

## 5. Conclusions

In conclusion, verum acupuncture, compared to sham or no-acupuncture groups, improved FC and peripheral muscle strength in patients with maintenance HD, and in some parameters, the results persist into the 12-week follow-up period. Although acupuncture treatment once a week appeared to produce similar effects compared to three times a week, more solid prospective research will be needed to validate the impact of the frequency of acupuncture treatment.

## Figures and Tables

**Figure 1 healthcare-10-01947-f001:**
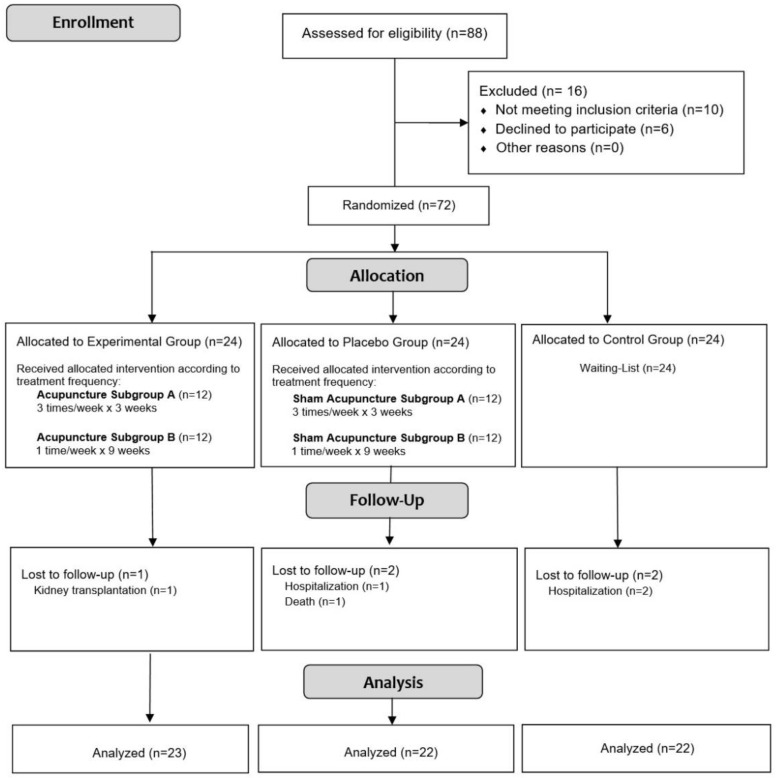
Study flow diagram.

**Figure 2 healthcare-10-01947-f002:**
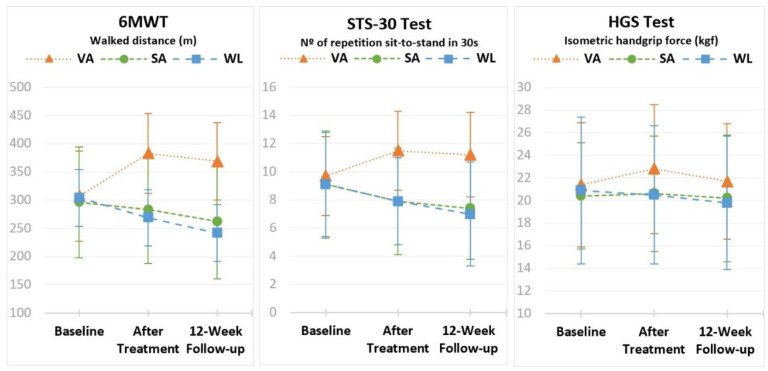
Means (±SD) of 6MWT, STS-30 Test and HGS Test in the baseline assessment, after treatment and follow-up assessment, within each group (VA—Verum Acupuncture Group, SA—Sham Acupuncture Group, WL—Waiting-List Group).

**Table 1 healthcare-10-01947-t001:** Details of needling and treatment regimen.

		Experimental Group		Placebo Group		Control Group
		Verum Acupuncture (VA)		Sham Acupuncture (SA)		Waiting-List (WL)
**DETAILS OF NEEDLING**	**Number of** **needles**	5 fixed acupoints and a total of 8 needle insertions per subject and session.		5 fixed acupoints and a total of 8 needle insertions per subject and session.		No acupuncture treatment will be performed from the time of randomization until the end of the follow-up period.
	**Names of** **points used**	*Taixi* (KI3), bilateral. *Sanyinjiao* (SP6), bilateral. *Zusanli* (ST36), bilateral. *Shenmen* (HT7) unilateral, in the arm without arteriovenous fistula. *Guanyuan* (CV4), unilateral.		Non-acupuncture points near the acupuncture points selected as described in Appendix B, Table A2.		
	**Depth** **of insertion**	CV4, ST36, KI3 and SP6 were inserted perpendicularly (15 to 20 mm depth) and HT7 was inserted slightly oblique (10 mm depth).		Superficial needling (5 mm depth) at non-acupuncture points abovementioned.		
	**Response sought**	*De qi* sensation (described as a compositional sensation including numbness, soreness, distention, heaviness) was achieved through lifting, thrusting and twirling manipulations.		No *De qi* sensation.		
	**Needle** **stimulation**	Manual stimulation. After generating needling sensation, needles were manipulated for one minute every ten minutes during needle retention.		No stimulation.		
	**Needle** **retention time**	25 min.		25 min.		
	**Needle type**	Sterilized stainless-steel needle (0.25 × 25 mm) Tewa, asia-med GmbH, Kirchplatz 182,049 Pullach, Germany.		Sterilized stainless-steel needle (0.25 × 25 mm) Tewa, asia-med GmbH, Kirchplatz 182,049 Pullach, Germany.		
**TREATMENT** **REGIMEN**	**Number of** **treatment** **sessions**	9 treatment sessions.		9 treatment sessions.		Not applicable.
	**Frequency** **of treatment** **sessions**	VA subgroup A (VA.A) Three times a week for 3 weeks. (3 × 3)	VA subgroup B (VA.B) Once a week for 9 weeks. (1 × 9)	SA subgroup A (SA.A) Three times a week for 3 weeks. (3 × 3)	SA subgroup B (SA.B) Once a week for 9 weeks. (1 × 9)	

**Table 2 healthcare-10-01947-t002:** Sociodemographic, clinical and laboratorial characteristics of the sample, overall and by group (baseline).

Variables	Total (*n* = 67)	Verum Acupuncture Group (*n* = 23)	Sham Acupuncture Group (*n* = 22)	Non-Acupuncture Group (*n* = 22)	*p*
**Sociodemographic**					
**Sex**					
Female	26 (38.8%)	9 (39.1%)	8 (36.4%)	9 (40.9%)	*1.000 **^(1)^***
Male	41 (61.2%)	14 (60.9%)	14 (63.6%)	13 (59.1%)	
**Age**					
Minimum–Maximum	56–91	60–84	57–91	56–87	
Mean (SD)	71.6 (7.7)	71.2 (5.1)	72.6 (8.3)	71.0 (9.4)	*0.764 **^(2)^***
**Residence**					
Urban	24 (35.8%)	9 (39.1%)	8 (36.4%)	7 (31.8%)	*0.948 **^(1)^***
Rural	43 (64.2%)	14 (60.9%)	14 (63.6%)	15 (68.2%)	
**Education level**					
No literacy	5 (7.5%)	0 (0.0%)	2 (9.1%)	3 (13.6%)	*0.279 **^(1)^***
1° Cycle (4 years)	49 (73.1%)	20 (87.0%)	17 (77.3%)	12 (54.5%)	
2° Cycle (6 years)	7 (10.4%)	2 (8.7%)	1 (4.5%)	4 (18.2%)	
3° Cycle (9 years)	1 (1.5%)	0 (0.0%)	0 (0.0%)	1 (4.5%)	
High school (12 years)	5 (7.5%)	1 (4.3%)	2 (9.1%)	2 (9.1%)	
**Professional status**					
Employed	4 (6.0%)	0 (0.0%)	2 (9.1%)	2 (9.1%)	*0.481 **^(1)^***
Self-employed	4 (6.0%)	2 (8.7%)	0 (0.0%)	2 (9.1%)	
Unemployed	2 (3.0%)	1 (4.3%)	1 (4.5%)	0 (0.0%)	
Retired	57 (85.1%)	20 (87.0%)	19 (86.4%)	18 (81.8%)	
**Clinical**					
**CKD causes**					
Diabetes mellitus	29 (43.3%)	10 (43.5%)	10 (45.5%)	9 (40.9%)	*0.537 **^(1)^***
Chronic rejection	7 (10.4%)	1 (4.3%)	4 (18.2%)	2 (9.1%)	
Hypertensive nephropathy	4 (6.0%)	3 (13.0%)	0 (0.0%)	1 (4.5%)	
High blood pressure	2 (3.0%)	0 (0.0%)	1 (4.5%)	1 (4.5%)	
Glomerulonephritis	2 (3.0%)	1 (4.3%)	0 (0.0%)	1 (4.5%)	
Polycystic disease	2 (3.0%)	0 (0.0%)	0 (0.0%)	2 (9.1%)	
Interstitial tubular necrosis	2 (3.0%)	0 (0.0%)	2 (9.1%)	0 (0.0%)	
Other	5 (7.5%)	3 (13.0%)	1 (4.5%)	1 (4.5%)	
Unknown	14 (20.9%)	5 (21.7%)	4 (18.2%)	5 (22.7%)	
**Hemodialysis time**					
<12 months	2 (3.0%)	2 (8.7%)	0 (0.0%)	0 (0.0%)	*0.240 **^(1)^***
12 to 120 months	56 (83.6%)	19 (82.6%)	20 (90.9%)	17 (77.3%)	
>120 months	9 (13.4%)	2 (8.7%)	2 (9.1%)	5 (22.7%)	
**Vascular access**					
Arteriovenous fistula	61 (91.0%)	22 (95.7%)	18 (81.8%)	21 (95.5%)	*0.306 **^(1)^***
Central venous catheter	6 (9.0%)	1 (4.3%)	4 (18.2%)	1 (4.5%)	
**Laboratorial**	Mean (SD)				
Hemoglobin	10.93 (1.03)	10.88 (0.94)	10.86 (1.05)	11.05 (1.12)	*0.797 **^(2)^***
Potassium	5.44 (0.81)	5.52 (0.89)	5.35 (0.81)	5.45 (0.74)	*0.799 **^(2)^***
Calcium	9.11 (0.51)	9.13 (0.55)	9.10 (0.49)	9.10 (0.51)	*0.970 **^(2)^***
Phosphorus	4.74 (1.13)	4.71 (1.17)	4.70 (1.25)	4.80 (1.00)	*0.944 **^(2)^***
Sodium	138.1 (3.0)	138.5 (2.9)	138.6 (3.8)	137.3 (2.0)	*0.296 **^(2)^***
Albumin	3.91 (0.28)	3.98 (0.27)	3.81 (0.27)	3.95 (0.29)	*0.101 **^(2)^***
Urea (before)	152.6 (37.5)	161.0 (37.9)	145.3 (36.2)	151.0 (38.3)	*0.369 **^(2)^***
Creatinine	9.94 (2.12)	10.15 (2.26)	9.51 (1.94)	10.16 (2.16)	*0.514 **^(2)^***
Parathyroid hormone	418.0 (243.2)	454.2 (256.9)	358.2 (193.3)	439.9 (271.5)	*0.370 **^(2)^***
Cholesterol	162.8 (37.1)	166.0 (47.2)	155.5 (35.8)	166.6 (25.0)	*0.536 **^(2)^***

(1) significance value of Fisher’s Exact Test; (2) significance value of ANOVA.

**Table 3 healthcare-10-01947-t003:** Results of the ANOVA (time*group*treatment frequency) for 6MWT, STS-30 Test and HGS Test.

Effects	6MWT	STS-30 Test	HGS Test
Time	*p* = 0.001, η^2^_p_ = 0.114	*p* < 0.001, η^2^_p_ = 0.240	*p* = 0.004, η^2^_p_ = 0.168
Interaction time*group	*p* < 0.001, η^2^_p_ = 0.521	*p* < 0.001, η^2^_p_ = 0.600	*p* = 0.032, η^2^_p_ = 0.083
Interaction time*treatment frequency	*p* = 0.172, η^2^_p_ = 0.028	*p* = 0.438, η^2^_p_ = 0.027	*p* = 0.843, η^2^_p_ = 0.002

*p*—*p*-value of the terms of the mixed ANOVA; η^2^p—partial Eta squared.

**Table 4 healthcare-10-01947-t004:** Descriptive statistics and comparison of 6-Minute Walk Test (6MWT), 30-Second Sit-to-Stand (STS-30) Test and Handgrip Strength (HGS) Test by group and through time.

	Verum Acupuncture (VA) Group (*n* = 23)	Sham Acupuncture (SA) Group (*n* = 22)	Waiting-List (WL) Group (*n* = 22)
	Mean ± SD	Mean ± SD	Mean ± SD
6MWT, Walked distance (m)			
Baseline	307.2 ± 80.1 ^a^	296.2 ± 97.8 ^a^	304.1 ± 119.3 ^a^
After treatment	383.2 ± 70.7 ^b^	283.4 ± 95.2 ^a^	268.8 ± 113.8 ^b^
Follow-up	368.9 ± 68.2 ^b^	262.5 ± 101.8 ^b^	241.7 ± 118.1 ^c^
*ANOVA ^(1)^*	*p < 0.001*	*p < 0.001*	*p < 0.001*
*Interaction time*group*	*p < 0.001, η^2^_p_ = 0.509*		
STS-30 Test, Repetition number of sit-to-stand in 30 s			
Baseline	9.7 ± 2.8 ^a^	9.1 ± 3.8 ^a^	9.1 ± 3.7 ^a^
After treatment	11.5 ± 2.8 ^b^	7.9 ± 3.8 ^b^	7.9 ± 3.1 ^b^
Follow-up	11.2 ± 3.0 ^b^	7.4 ± 3.6 ^b^	7.0 ± 3.7 ^c^
*ANOVA ^(1)^*	*p < 0.001*	*p < 0.001*	*p < 0.001*
*Interaction time*group*	*p < 0.001, η^2^_p_ = 0.565*		
HGS Test, Isometric handgrip force (kgf)			
Baseline	21.4 ± 5.5 ^a^	20.4 ± 4.7 ^a^	20.9 ± 6.5 ^a^
After treatment	22.8 ± 5.7 ^b^	20.6 ± 5.1 ^a^	20.5 ± 6.1 ^a^
Follow-up	21.7 ± 5.1 ^ab^	20.2 ± 5.6 ^a^	19.8 ± 5.9 ^a^
*ANOVA ^(1)^*	*p = 0.012*	*p = 0.601*	*p = 0.112*
*Interaction time*group*	*p = 0.043, η^2^_p_ = 0.083*		

(1) *p*-value of repeated measures ANOVA for comparison through time within each group; ^a^, ^b^, ^c^ moments (baseline, after treatment, follow-up) with a superscript letter in common do not differ significantly: *p* > 0.05 in the multiple comparison tests with Bonferroni correction; η^2^p—partial Eta squared.

**Table 5 healthcare-10-01947-t005:** Assessment of blinding success.

	Participants’ Guess, *n* (%)		
Groups	Verum Acupuncture	Sham Acupuncture	Do Not Know
Verum Acupuncture (*n* = 23)	7 (30.4%)	0 (0.0%)	16 (69.6%)
Sham Acupuncture (*n* = 22)	7 (31.8%)	0 (0.0%)	15 (68.2%)
Total (*n* = 45)	14 (31.1%)	0 (0.0%)	31 (68.9%)
Blinding Index: 0.84 (95% Confidence Interval: 0.78–0.91)			

## Data Availability

Not applicable.

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
