# Peer review of "Effect of Acupuncture on Functional Capacity in Patients Undergoing Hemodialysis: A Patient-Assessor Blinded Randomized Controlled Trial"

_healthcare, 2022, doi:10.3390/healthcare10101947_

Round 1
Reviewer 1 Report
The present study was designed to evaluate the effect of acupuncture on FC in patients with CKD G5 undergoing HD and to analyze the differences in the frequency of treatment (three times a week or once a week). Considering a lack of similar studies in current literature, I believe the findings of this study can provide some merits and contributions in this research field. On the whole, this study is designed with sound methodology; the manuscript is well written and reported in a proper way, but I raise several concerns as follows.
(1) Title and main body: the wording of “single-blinded” is amphibolous, please consider using “patient blinded” or “patient-assessor blinded”.
(2) Since patients were blinded in this trial, was the success of patient blinding evaluated at specific time points during the trial? Was the blinding index calculated?
(3) The sample calculation method needs more details. How can you determine that “A minimum of 20 patients in each group was estimated to achieve a small to medium effect size (f = 0.19) [39]”? The sample size is generally calculated based on specific reference data of the primary outcome, which can be gained from pilot data or published data of previous studies. What is the source of your reference data to calculate the sample size? And please provide the computational formula of sample size calculation if applicable.
(4) Given that your primary interest is the effect of acupuncture on patients’ functional capacity (FC), kindly explain why Shenmen (HT7) and Guan Yuan (CV4) are used as two of the critical acupoints in your study in the Discussion section. If possible, provide references for the statement that “The selection of acupuncture points was based on the literature review, on the general principles of acupuncture and Traditional Chinese Medicine (TCM) meridian theory........acupuncture and nephrology”.
(5) According to the record on Clinicaltrials.gov, you retrospectively registered this trial. Please provide the reason why it is not prospectively registered to avoid the risk of bias as much as possible. Moreover, the outcomes reported in this manuscript are not totally consistent with the planned outcomes listed on the Clinicaltrials.gov website. Health-Related Quality of Life outcomes are supposed to be one of the primary outcomes when you registered this study on Clinicaltrials.gov, but it appears that they are not reported in the current paper.
Author Response
Manuscript healthcare-1922421
Response to Reviewer 1 Comments
First of all, we thank the referee for the time spent carefully reviewing our manuscript and its opinions regarding the science and presentation of the material. In what follows, the referee's comments are in black and the author's responses are in red.
Point 1: Title and main body: the wording of “single-blinded” is amphibolous, please consider using “patient blinded” or “patient-assessor blinded”.
Response 1: We agree and have updated the wording “single-blinded” to “patient-assessor blinded” in the title and main body.
Point 2: Since patients were blinded in this trial, was the success of patient blinding evaluated at specific time points during the trial? Was the blinding index calculated?
Response 2: After the treatment, we applied a questionnaire in wish we asked the participants of Verum Acupuncture (VA) and Sham Acupuncture (SA) groups what type of acupuncture treatment (verum or sham) they believe they had received. This question allowed us to calculate the Blinding Index – information and results added in sections “2.7 Statistical analysis and sample size” and “3. Results”.
Point 3: The sample calculation method needs more details. How can you determine that “A minimum of 20 patients in each group was estimated to achieve a small to medium effect size (f = 0.19) [39]”? The sample size is generally calculated based on specific reference data of the primary outcome, which can be gained from pilot data or published data of previous studies. What is the source of your reference data to calculate the sample size? And please provide the computational formula of sample size calculation if applicable.
Response 3: No previous data were available for the sample size estimation due to the lack of similar studies in current literature and the absence of a pilot study. As we did not have previous information, the sample size was estimated based on the statistical test planned (mixed ANOVA), considering the usual values of statistical power (80%) and significant level (5%) to achieve a small/medium effect size – considering the thresholds proposed by Cohen (cited in the paper).
The computational formulas can be found in the cited paper that describes the software used for the sample size estimation – “G*Power 3: A flexible statistical power analysis program for the social, behavioral, and biomedical sciences” (Faul F, Erdfelder E, Lang AG, Buchner A. G*Power 3: a flexible statistical power analysis program for the social, behavioral, 554 and biomedical sciences. Behav Res Methods. 2007;39(2):175-91. doi:10.3758/bf03193146).
The sample size description was reformulated.
Point 4: Given that your primary interest is the effect of acupuncture on patients’ functional capacity (FC), kindly explain why Shenmen (HT7) and Guan Yuan (CV4) are used as two of the critical acupoints in your study in the Discussion section.
If possible, provide references for the statement that “The selection of acupuncture points was based on the literature review, on the general principles of acupuncture and Traditional Chinese Medicine (TCM) meridian theory........acupuncture and nephrology”.
Response 4: Patients undergoing hemodialysis have to cope with a stressful and disruptive chronic illness and suffer from emotional problems that negatively impact on their overall health. Given this assumption and the fundamentals of TCM, the acupuncture points Shenmen (HT7) and Guan Yuan (CV4) were included in the study protocol. The Shenmen (HT7) acupoint was chosen since it is an important acupuncture point for calming and regulating the spirit. The CV4 acupuncture point is described as a crucial point for strengthening the body and mind. Moreover, as it tonifies the kidney and the original Qi, it is recognized as an effective point in the case of chronic diseases. This explanation was added to the Discussion section.
References for the statement “The selection of acupuncture points was based on the literature review, on the general principles of acupuncture and Traditional Chinese Medicine (TCM) meridian theory........acupuncture and nephrology” were added.
Point 5: According to the record on Clinicaltrials.gov, you retrospectively registered this trial. Please provide the reason why it is not prospectively registered to avoid the risk of bias as much as possible.
Moreover, the outcomes reported in this manuscript are not totally consistent with the planned outcomes listed on the Clinicaltrials.gov website. Health-Related Quality of Life outcomes are supposed to be one of the primary outcomes when you registered this study on Clinicaltrials.gov, but it appears that they are not reported in the current paper.
Response 5: As mentioned in our paper, this study is an integral part of the research project of the Doctoral Program in Biomedical Sciences of ICBAS School of Medicine and Biomedical Sciences, University of Porto, Portugal. The beginning of our study occurred after the approval of the Ethics Committee of Centro Hospitalar Universitário do Porto/Instituto Ciências Biomédicas Abel Salazar (registered number 2019/CE/P026_P304/2019/CETI). However, due to a time lapse between the research team's proposal and the registration procedure associated with our university, the registration of the present study on ClinicalTrials.gov was unfortunately done retrospectively.
Regarding the Health-Related Quality of Life outcomes, only data at the different evaluation times were collected. The data are currently being statistically analyzed and another paper entitled "Effectiveness of acupuncture versus sham acupuncture on health-related quality of life of hemodialysis patients: a patient-assessor blinded randomized controlled trial" is being drafted.
Once more, we would like to thank the referee for taking the time to review our manuscript.
Marta Raquel Custódio Correia de Carvalho
(Corresponding Author)

Reviewer 2 Report
I would like to thank the Journal of Healthcare and the editor-in-chief for inviting us to review this interesting manuscript.
After an in-depth review of this manuscript (Manuscript ID: healthcare-1922421), I have compiled my opinions and suggestions. Please see below.
# Overall opinion
It is an RCT that is thoughtfully well designed and performed without significant flaws. Since the manuscript has sufficient scientific completeness and novelty, I think it is a good source of evidence that can be helpful to actual patients and clinicians. Publishing in the ‘Healthcare’ journal does not appear to be a problem after corrections for the following several concerns.
# Major issues
1. Conclusion
I think the overall findings of this study are meaningful. However, considering the sample size and study design for each group, the conclusion that “acupuncture treatment once a week produced similar effects to treatment three times a week” is difficult to support only with the design and results in this RCT. Therefore, even if the authors' opinions are sufficiently expressed in the discussion, in conclusion, it seems appropriate to relax the expression to the extent that it is necessary to additionally verify the problem of the treatment session through a separate study in the future.# Minor issues
# minor issues
1. Introduction
It would be better if the introduction was concisely reduced to 2-3 paragraphs. In particular, information on CKD does not need to be provided in too much detail, and in the case of citations to previous studies, it is thought that it is sufficient to mention only two of the most important and up-to-date.
2. Method
Please describe the allocation ratio for each group explicitly in the method section. Also, it would be nice if you could add an explanation as to why the allocation ratio was designed that way.
3. References
Although the research topic of this manuscript is very novel, many more recent studies can support it. Consider citing more references published between 2020-2022 throughout the manuscript.
I hope that my comments above will help improve and publish the manuscript.
Author Response
Manuscript healthcare-1922421
Response to Reviewer 2 Comments
First of all, we thank the referee for the time spent carefully reviewing our manuscript and its opinions regarding the science and presentation of the material. In what follows, the referee's comments are in black and the author's responses are in red.
# Major issues
Point 1: Conclusion
I think the overall findings of this study are meaningful. However, considering the sample size and study design for each group, the conclusion that “acupuncture treatment once a week produced similar effects to treatment three times a week” is difficult to support only with the design and results in this RCT. Therefore, even if the authors' opinions are sufficiently expressed in the discussion, in conclusion, it seems appropriate to relax the expression to the extent that it is necessary to additionally verify the problem of the treatment session through a separate study in the future
Response 1: We appreciate the reviewer’s insightful suggestion and the Conclusion was reformulated.
# Minor issues
Point 1: Introduction
It would be better if the introduction was concisely reduced to 2-3 paragraphs. In particular, information on CKD does not need to be provided in too much detail, and in the case of citations to previous studies, it is thought that it is sufficient to mention only two of the most important and up-to-date.
Response 1: We appreciate the reviewer’s suggestion and the Introduction was reduced.
Point 2: Method
Please describe the allocation ratio for each group explicitly in the method section. Also, it would be nice if you could add an explanation as to why the allocation ratio was designed that way.
Response 2: We apologize if we missed the allocation ratio description. The method section was reformulated with that.
Point 3: References
Although the research topic of this manuscript is very novel, many more recent studies can support it. Consider citing more references published between 2020-2022 throughout the manuscript.
Response 3: More recent references were added.
Once more, we would like to thank the referee for taking the time to review our manuscript.
Marta Raquel Custódio Correia de Carvalho
(Corresponding Author)
